# Deep Relational Factorization Machines

## Abstract

Factorization Machines (FMs) is an important supervised learning approach due to its unique ability to capture feature interactions when dealing with high-dimensional sparse data. However, FMs assume each sample is independently observed and hence incapable of exploiting the interactions among samples. On the contrary, Graph Neural Networks (GNNs) has become increasingly popular due to its strength at capturing the dependencies among samples. But unfortunately, it cannot efficiently handle high-dimensional sparse data, which is quite common in modern machine learning tasks. In this work, to leverage their complementary advantages and yet overcome their issues, we proposed a novel approach, namely Deep Relational Factorization Machines, which can capture both the feature interaction and the sample interaction. In particular, we disclosed the relationship between the feature interaction and the graph, which opens a brand new avenue to deal with high-dimensional features. Finally, we demonstrate the effectiveness of the proposed approach with experiments on several real-world datasets.

## 1 Introduction

Many supervised learning tasks need to model data with numerous categorical features, which is usually converted into a set of binary features using one-hot encoding. However, when the original categorical features have high cardinalities, such data becomes high-dimensional and sparse. The difficulty in modeling such data is that, most machine learning techniques rely on co-occurrence of features to model their interactions, while in sparse data such co-occurrences are relatively rare compared to the number of possible feature combinations, and hence over-fitting occurs. This is particularly common in web applications, which may involve high-cardinality categorical features such as user IDs, item IDs, and ZipCodes, etc. Factorization Machines (FMs) was introduced byRendle (2010) to model such high-dimensional sparse data. The key idea of FMs is learning a latent vector of each one-hot encoded feature, and capture an arbitrary pairwise (order-2) interaction by inner product of respective latent vectors. The success of FMs has been evidenced by applications such as click-through rate prediction (Rendle, 2012; Ta, 2015) and recommendation (Rendle et al., 2011; Wu et al., 2017).

To further improve the performance of FMs, numerous variants have been proposed (Blondel et al., 2016; Chen et al., 2019; Guo et al., 2017; He & Chua, 2017; Juan et al., 2016; Lian et al., 2018; Qu et al., 2016; Wang et al., 2017; Xiao et al., 2017). For instance, Field-aware Factorization Machine (FFM) (Juan et al., 2016) is proposed to conduct fine-grained feature interaction. With the development of deep neural networks in recent years, some deep variants have been proposed. For instance, DeepFM (Guo et al., 2017) combines FM and deep neural networks (DNN) to do both the second-order and high-order feature interaction. Deep & Cross Network (DCN) (Wang et al., 2017) stacks multiple interaction layer to learn the high-order feature interaction. Attentional Factorization Machines (Xiao et al., 2017) employs a neural attention network to additionally weight each interaction term in FMs. Order-aware Embedding Neural Network (Guo et al., 2019) learns an order-specific latent vector for each binary feature in Higher-order Factorization Machines.

However, all the aforementioned FMs variants only focus on the feature interaction. In real-world applications, there also exists sample interaction. For instance, when predicting CTR in a social network, two users in the same basketball group have a large probability to click the same advertisement of basketball shoes since they are supposed to have similar hobbies for basketball. Thus, it is necessary and helpful to incorporate the sample interaction when conducting prediction. Graph Convolutional Networks (GCN) and the ilk (Kipf & Welling, 2016) use a graph convolution opera-

tion to capture the correlation between nodes in a graph. Specifically, this operation aggregates all neighbors' information when making prediction for each node (sample) in a graph. As a result, the prediction encodes the interaction between nodes. Recently, GCN has been widely used in numerous applications to capture the interaction in the sample level, such as recommendation (Wang et al., 2019; Ying et al., 2018). However, although GCN can incorporate the sample interaction, yet it is difficult for GCN to deal with the sparse categorical features well.

Summarily, FMs and GCNs are two general classes of techniques that are typically used for different applications. Both have been shown to be the state-of-the-art in their own respective areas. Meanwhile, they also suffer their intrinsic drawbacks. To overcome the disadvantages of using either approach independently while inheriting the complementary advantages of both approaches, a straightforward solution is to combine these two technologies together. However, how to seamlessly unify these two independent models together to capture both the feature interaction and sample interaction is challenging. To address this issue, we propose a novel Deep Relational Factorization Machine (DRFM). In particular, to model the relationship between different features, we tackle it from the perspective of graph. More specifically, the interaction in different orders between features is reformulated by the path in a feature concurrence graph, with which our method can easily capture the feature interaction in different order by using graph convolutional operation. As far as we know, this is the first work dealing with the feature interaction from the graph view. Moreover, to model the interaction between different samples, we proposed a novel sample interaction component which can capture the high-order sample interaction both linearly and exponentially. Extensive experimental results confirmed the effectiveness of our proposed method. At last, we summarize the contribution of this work as follows:

- We disclosed the relationship between the feature interaction and the graph, and proposed a novel graph-based method to deal with the feature interaction. This opens a new avenue to deal with high-dimensional categorical features.

- We proposed a general framework that fuses FMs and GCNs into a single unified learning approach, called DRFM. It overcomes the disadvantages of using either approach independently while inheriting the complementary advantages of both approaches.

- We demonstrated the effectiveness of DRFM for both link prediction and regression tasks.

## 2 RELATED WORK

Factorization Machines (FM) was first proposed by Rendle (2010). With a factorized interaction term, FM is good at dealing with the data with sparse categorical features. However, the standard FM can only capture the second-order feature interaction. To utilize the high-order feature interaction, Blondel et al. (2016) proposed the high-order factorization machine (HOFM) which explicitly incorporates the high-order feature combinations. In addition, to conduct fine-grained feature interaction, Juan et al. (2016) proposed the Field-aware Factorization Machine (FwFM) which assigns multiple latent representation to each feature in terms of feature groups. Similarly, Chen et al. (2019) also proposed to represent each feature with multiple latent representations according to the frequency of feature occurrences.

Recently, deep neural networks (DNN) have shown promising performance on a wide variety of tasks, such as computer vision, natural language processing. Inspired by this, some researchers proposed to combine DNN with FM to fully utilize their advantages. For instance, Factorization-machine supported Neural Networks (FNN) first pre-trains the factorization machine to get the latent representation of features and then feeds these representations to DNN to learn high-order feature interaction implicitly (Zhang et al., 2016). To train FM and DNN in an end-to-end way, Product-based Neural Network (PNN) proposed by Qu et al. (2016) introduced a product layer to connect the feature embedding layer and DNN layers. However, both of these two models only focus on the high-order feature interaction, ignoring the low-order interaction. To address this issue, Guo et al. (2017) proposed DeepFM which models FM and DNN in two branches and trains them simultaneously. Wang et al. (2017) proposed Deep & Cross Network (DCN) to explicitly capture the feature interaction with different orders. Similarly, xDeepFM (Lian et al., 2018) also aims at capturing feature interactions with different orders, but it uses the inner product rather than outer product like DCN.

As we discussed earlier, FM and its variants aim at capturing the feature interaction. In some real-world applications, it is necessary to capture the sample interaction. To do that, Kipf & Welling (2016) proposed graph convolutional networks (GCN). Specifically, GCN employs the graph convolutional operation to capture the interaction between samples and their neighbors. Recently, GCN has been applied to various tasks to capture the sample interaction. For instance, Ying et al. (2018) proposed PinSage to explore the item-item interaction in the recommender system. Wang et al. (2019) proposed the Neural Graph Collaborative Filtering (NGCF) to utilize the user-item interaction for recommendation.

## 3 PRELIMINARIES

In this section, we are going to present some preliminaries about factorization machine and graph convolutional neural networks.

Throughout this paper, a graph is represented by $\mathcal{G} = (\mathcal{V}, \mathcal{E})$ where $\mathcal{V} = \{v_i\}$ represents the node set and $\mathcal{E} = \{(v_i, v_j)\}$ represents the edge set. In this paper, we focus on the attributed graph and the node feature matrix is represented by $\mathbf{X} = [\mathbf{x}_1, \mathbf{x}_2, \cdots, \mathbf{x}_n] \in \mathbb{R}^{d \times n}$. In this work, we focus on the high-dimensional categorical features, which are very common in real-world applications, such as recommendation and CTR prediction. Specifically, we assume the feature of each node is represented as $\mathbf{x}_i = [0, 1, 0, \cdots, 1, 1, 0, 0]^T \in \mathbb{R}^d$ whose features is categorical and the number of non-zero values is much less than $d$. In addition, if a node has the ground-truth (e.g. the regression task in our experiment), it is denoted by $\mathbf{Y} = [y_i] \in \mathbb{R}^n$. Note that we will use samples and nodes interchangeably throughout this paper.

Based on aforementioned terminologies, Factorization Machine (FM) is defined as follows:

$$\hat{y}_i = b + \mathbf{w}^T \mathbf{x}_i + \sum_{p < q} \langle \mathbf{v}_p, \mathbf{v}_q \rangle x_{i,p} x_{i,q} \tag{1}$$

where $\hat{y}_i \in \mathbb{R}$ denotes the prediction of node $v_i$, $\mathbf{x}_i = [x_{i,p}] \in \mathbb{R}^d$ represents node features where $x_{i,p}$ is the $p$-th feature of node $v_i$, $\mathbf{v}_p \in \mathbb{R}^k$ stands for the embedding of the $p$-th feature. Compared the regular linear model, FM can capture the interaction between different features. Specifically, in the non-linear term, the dot product $\langle \mathbf{v}_p, \mathbf{v}_q \rangle$ computes the interaction between feature $x_{i,p}$ and $x_{i,q}$. However, FM can only capture the interaction inside each node, ignoring interaction between different nodes.

Convolution is an effective operator to capture the local correlation. The regular convolutional operator is used to extract features by exploring the feature correlation. On the contrary, the graph convolutional operator is proposed to explore the sample correlation. Specifically, the graph convolutional operation in the $l$-th hidden layer of the Graph Convolutional Neural Network (GCN)is defined as follows:

$$\mathbf{z}_i^{l+1} = \frac{1}{\sqrt{|\mathcal{N}(i)|}} \sum_{i' \in \mathcal{N}(i)} \frac{1}{\sqrt{|\mathcal{N}(i')|}} \mathbf{W}^{l+1} \mathbf{h}_{i'}^l$$
$$\mathbf{h}_i^{l+1} = f(\mathbf{z}_i^{l+1}) \tag{2}$$

where $\mathbf{h}_i^l \in \mathbb{R}^{d_l}$ denotes the hidden representation of node $v_i$ in the $l$-th layer, $\mathbf{W}^{l+1} \in \mathbb{R}^{d_l \times d_{l+1}}$ represents the model parameter, $\mathcal{N}(i)$ indicates the neighbors of node $v_i$, and $f(\cdot)$ stands for the non-linear activation function. It can be found that the representation $\mathbf{z}_i^{l+1}$ of the $i$-th sample is constructed by aggregating its neighbors. In this way, GCN can capture the sample interaction. However, although GCN can explore the interaction between different samples, yet it is not good at exploring the feature interaction. In the following section, we will propose a new model to address these two issues of FM and GCN.

## 4 DEEP RELATIONAL FACTORIZATION MACHINE

As shown in Eq. (1), the regular FM (Rendle, 2010) only considers the interaction between different features, ignoring the interaction between different samples. In many real-world applications, the sample interaction might be important for prediction. For instance, in a recommender system, the interaction between users and items is important to make accurate recommendation. If ignoring this

kind of interaction, it will be difficult to find the potential items to do recommendation. Another good example is the campaign performance prediction in online advertising. Specifically, an advertiser may only change a small part of its original campaign, such as targeting countries, and launch a new campaign. As a result, these two campaigns share a lot of common information. If we can capture the correlation between these two campaigns, it will be beneficial to get better prediction result.

Based on the aforementioned intuition, a natural question is that: how to capture the feature interaction and sample interaction simultaneously? A straightforward method is to combine them together directly as follows:

$$
\begin{aligned}
y_i &= y_i^{FM} + y_i^{GCN} , \\
y_i^{FM} &= \mathbf{w}^T \mathbf{x}_i + \sum_{p<q} \langle \mathbf{v}_p, \mathbf{v}_q \rangle x_{i,p} x_{i,q} , \\
y_i^{GCN} &= g(f(\frac{1}{\sqrt{|\mathcal{N}(i)|}} \sum_{i' \in \mathcal{N}(i)} \frac{1}{\sqrt{|\mathcal{N}(i')|}} \mathbf{W}^{l+1} \mathbf{x}_{i'}^l))
\end{aligned}
\tag{3}
$$

where $g(\cdot)$ denotes the prediction function based on node features. Although this straightforward method can explore the feature interaction and sample interaction simultaneously, yet it can be seen that GCN and FM are almost independent. Specifically, the prediction from FM does not use the sample interaction and that from GCN does not involve the feature interaction too.

To address this issue, we propose the Deep Relational Factorization Machine (DRFM). In detail, our proposed DRFM also has two components: the sample interaction component and the relational feature interaction component. As for the sample interaction component, we proposed a novel sample interaction layer which acts on the *sample graph*. As for the relational feature interaction component, we proposed to capture both the high-order feature interaction based on the *feature graph* and the sample interaction based on the *sample graph*.

## 4.1 RELATIONAL FEATURE INTERACTION

The relational feature interaction (RFI) component aims at dealing with the categorical features to capture the feature interaction. At the same time, it should capture the sample interaction. Based on these goals, the prototype of our relational feature interaction component is defined as follows:

$$
\begin{aligned}
h^{FI}(\mathbf{x}_i) &= \mathbf{w}^T \mathbf{x}_i + \sum_{p<q} \langle \mathbf{v}_p, \mathbf{v}_q \rangle x_{i,p} x_{i,q} \\
y_i^{RFI} &= \frac{1}{\sqrt{|\mathcal{N}(i)|}} \sum_{i' \in \mathcal{N}(i)} \frac{1}{\sqrt{|\mathcal{N}(i')|}} h^{FI}(\mathbf{x}_{i'})
\end{aligned}
\tag{4}
$$

where $h^{FM}(\mathbf{x}_{i'})$ denotes the feature-interaction kernel which is used to deal with categorical features to capture the feature interaction, $y_i^{RFI}$ represents the prediction for node $v_i$ from the RFI component which considers both the feature interaction and sample interaction. Compared with the naive method, we can find that our proposed RFI can capture the sample interaction when dealing with categorical features, while the naive method cannot.

However, Eq. (4) can only capture the second-order interaction between different features, ignoring the high-order interaction. As we know, second order may be not enough due to the complexity of real-world datasets. Thus, it is necessary to capture the high-order feature interaction. In addition, the relationship between different features might be highly non-linear. Thus, it is important to explore the non-linearity between different features. To address these issues, we further propose a novel high-order feature-interaction kernel. Specifically, we deal with this issue from a totally new perspective. In particular, given a sample with categorical features, we can construct a concurrent feature graph in terms of the concurrence between different features. For instance, given $\mathbf{x} = [0, 1, 0, 1, 1, 0, 0]$ where the first, fourth, and fifth feature appear simultaneously, then there should be a link between them in the concurrence graph due to their concurrence, which is shown as

follows:

$$G = \begin{bmatrix} 1 & 0 & 0 & 0 & 0 & 0 & 0 \\ 0 & 1 & 0 & 1 & 1 & 0 & 0 \\ 0 & 0 & 1 & 0 & 0 & 0 & 0 \\ 0 & 1 & 0 & 1 & 1 & 0 & 0 \\ 0 & 1 & 0 & 1 & 1 & 0 & 0 \\ 0 & 0 & 0 & 0 & 0 & 1 & 0 \\ 0 & 0 & 0 & 0 & 0 & 0 & 1 \end{bmatrix} \tag{5}$$

For the concurrence graph, each feature is viewed as a node in the graph. As a result, a path in this graph indicates the concurrence of the features in this graph. Then, a long path corresponds to a high-order feature interaction, while a short path corresponds to a low-order feature interaction. Consequently, a lot of operations on the graph can be used to deal with the high-dimensional categorical features. In particular, inspired by the graph convolutional operation, we propose the following model to capture the high-order interaction between different features layer by layer:

$$
\begin{aligned}
\mathbf{v}_p^{l+1} &= graph\_conv(\mathbf{v}_p^0, \mathbf{v}_q^l) \\
\mathbf{v}_p^0 &= \sigma(W\mathbf{v}_p^0) \\
\mathbf{v}_p^{l+1} &= \sigma(W\mathbf{v}_p^{l+1}) \\
\mathbf{h}_i^{l+1} &= \sum_{p:x_{i,p}=1} \mathbf{v}_p^{l+1}
\end{aligned}
\tag{6}
$$

where $\mathbf{v}_p^l$ denotes the embedding of the $p$-th feature in the $l$-th layer. $\mathbf{v}_p^l$ encodes the high-order interaction in high layers. Here, we use the $graph\_conv$ to capture the interaction between different features. Unlike the standard graph convolutional operation, we propose the following interaction kernel:

$$graph\_conv(\mathbf{v}_p^0, \mathbf{v}_q^l) = \mathbf{v}_p^0 \circ \sum_{q:G_{pq}=1} \mathbf{v}_q^l \tag{7}$$

where $\circ$ denotes the element-wise product. It can be seen that we always use the embedding in the input layer $\mathbf{v}_p^0$ to interact with that in the high layer. In this way, the first layer will capture the second-order interaction. In the second layer, it will capture the third-order interaction, and in the high layers, high-order interaction will be captured. In other words, our method can capture the high-order interaction linearly. On the contrary, if we use $\mathbf{v}_p^l$ to do the product, the order will increase in an exponential way which might be too aggressive. Moreover, unlike existing high-order methods (Wang et al., 2017; Lian et al., 2018), we conduct the non-linear transformation for the feature embedding in each layer, which is shown in the second and third formulation in Eq. (6), to handle the highly non-linear relationship. At last, to capture the feature interaction in different order, we concatenate $\mathbf{h}_i^{l+1}$ in all layers as follows:

$$
\begin{aligned}
\mathbf{h}_i^{FI} &= [(\mathbf{h}_i^1)^T, (\mathbf{h}_i^2)^T, \cdots, (\mathbf{h}_i^L)^T]^T \\
\mathbf{h}_i^{RFI} &= \frac{1}{\sqrt{|\mathcal{N}(i)|}} \sum_{i' \in \mathcal{N}(i)} \frac{1}{\sqrt{|\mathcal{N}(i')|}} \mathbf{h}_{i'}^{FI}
\end{aligned}
\tag{8}
$$

In summary, we proposed a novel feature interaction component based on the feature concurrence graph, which can capture the high-order interaction and explore the non-linearity. As far as we know, this is the first work trying to deal with feature interaction from the graph perspective. This will open a new avenue to tackle high-dimensional categorical features by using graph operations.

## 4.2 SAMPLE INTERACTION

With our proposed RFI component, our model can capture the high-order feature interaction. However, the RFI component cannot capture high-order sample interaction. To address this issue, we develop a novel sample-interaction (SI) component, which is used to further explore the interaction between different samples. Moreover, the sample-interaction component should benefit from the RFI component. Therefore, we enforce the SI component shares the same feature embedding with RFI component. In this way, the feature embedding will be updated by both components. On the contrary, in the naive method, two components only share the raw input features. The high-order

sample interaction information cannot be used to update the feature embedding. As a result, in our method, the two components can benefit each other.

Specifically, the SI component is defined as follows:

$$\hat{\mathbf{h}}_i^l = \mathbf{h}_i^l + \frac{1}{\sqrt{|\mathcal{N}(i)|}} \sum_{i' \in \mathcal{N}(i)} \frac{1}{\sqrt{|\mathcal{N}(i')|}} \mathbf{h}_{i'}^l \circ \mathbf{h}_i^l$$

$$\mathbf{h}_i^{l+1} = \sigma(\mathbf{W}^{l+1} \hat{\mathbf{h}}_i^l) \tag{9}$$

where $\mathbf{h}_i^0 = \sum_{p:x_{i,p}=1} \mathbf{v}_p$. Compared with the regular graph convolutional operation, our method conducts an explicit sample interaction by $\mathbf{h}_{i'}^l \circ \mathbf{h}_i^l$. In addition, by using a residual connection, our method can capture the sample interaction both linearly and exponentially.

Similar with the feature interaction component, to get different orders of sample interaction, we keep all the intermediate $\mathbf{h}_i^l$. Then, we concatenate all of them as the node representation:

$$\mathbf{h}_i^{SI} = [(\mathbf{h}_i^1)^T, (\mathbf{h}_i^2)^T, \cdots, (\mathbf{h}_i^L)^T]^T \tag{10}$$

At last, after obtaining the representation from these two components, we concatenate them together for prediction as follows:

$$\hat{y} = [(\mathbf{h}_i^{RFI})^T, (\mathbf{h}_i^{SI})^T]W \tag{11}$$

With this prediction, we can use our model for both classification and regression tasks.

## 5 EXPERIMENTS

In this section, we design experiments to verify the performance of the proposed approach.

### 5.1 DATASETS

- Modcloth is a dataset where users rate the clothes they bought. Each user has five attributes: *[user_id, bra_size, cup_size, hips, height]*. Each cloth has three attributes: *[item_id, size, category]*. There are 47,185 users and 1,364 clothes.

- Renttherunway is a dataset where 88,178 users rate 5,795 clothes they rented. Here, each user has six attributes: *[user_id, weight, body_type, age, bust_size, height]* while each cloth has three attributes : *[item_id, size, category]*.

- Book-crossing contains the historical rating information for books by users. Users have three attributes: *[user_id, location, age]*. Books also have three attributes *[isbn, yearofpublication, publisher]*. In addition, the number of users is 278,858 and the number of books is 271,360.

- Company-X-CTR data is an advertiser-level CTR data which includes the winning rate of each campaign bidding for exposure. Each campaign has 63 categorical attributes.

In our experiments, all attributes of these datasets are transformed to categorical features. Then, we use the one-hot encoding to represent the categorical feature. Moreover, the first three datasets construct a bipartite graph respectively. As for the last dataset, campaigns construct a regular graph. At last, we summarize the statistics of these datasets in Table 1.

### 5.2 EXPERIMENTAL SETTINGS

Throughout our experiments, we evaluate our method on two tasks: *link prediction* (Section 5.3) and *regression* (Section 5.4). For the link prediction task, we use the first four datasets. In particular, all the existing links in a graph are viewed as positive links while non-existing links are treated negative ones. We randomly select 10% positive links for the training set and 10% positive links for the validation set. The rest positive links are used for the testing set. In addition, we randomly select the negative links for these three sets where the amount of negative links is same as that of positive links in each set. As for the regression task, we use the Company-X-CTR data, campaigns with the same

Table 1: Real-world data used for link prediction and regression. Here, the features of first three datasets include those of users and items.

| Dataset | #Nodes | #Edges | #Features |
|---|---|---|---|
| Modcloth | 48,549 | 40,607 | 47,222+1,395 |
| Renttherunway | 93,973 | 82,415 | 88,382+5,902 |
| Book-crossing | 550,218 | 515,568 | 336,283+288,268 |
| Company-X-CTR | 144,876 | 72,438 | 48,474 |

Table 2: Link prediction results (AUC).

| Methods | Modcloth | Renttherunway | Book-crossing |
|---|---|---|---|
| DeepFM | 0.8839 ($\pm$0.0230) | 0.7518 ($\pm$0.0236) | 0.6807 ($\pm$0.0651) |
| DCN | 0.8454 ($\pm$0.0640) | 0.6892 ($\pm$0.0467) | 0.6720 ($\pm$0.0321) |
| PNN | 0.9079 ($\pm$0.0083) | 0.7565 ($\pm$0.0221) | 0.7793 ($\pm$0.0344) |
| xDeepFM | 0.8919 ($\pm$0.0368) | 0.7314 ($\pm$0.0449) | 0.6917 ($\pm$0.0266) |
| GCN | 0.8985 ($\pm$0.0007) | 0.7428 ($\pm$0.0005) | 0.9075($\pm$0.0004) |
| DRFM | **0.9223** ($\pm$0.0003) | **0.7930** ($\pm$0.0015) | 0.9115($\pm$0.0005) |
| %Gain | **1.6%** | **4.9%** | **0.4%** |

placement id are connected to construct the graph. The winning rate of each campaign is normalized to $[0, 1]$.

To evaluate the performance of our proposed method, we compare it with 5 baseline methods, which are described as follows:

- DeepFM (Guo et al., 2017) combines the standard FM and multi-layer perceptron neural (MLP) network together where the input of MLP is the feature embedding from FM. DeepFM can capture the second-order feature interaction explicitly by FM and high-order feature interaction implicitly by MLP.

- DCN (Wang et al., 2017) is proposed to capture the high-order feature interaction explicitly by stacking multiple interaction layers together.

- PNN (Qu et al., 2016) uses a product layer to capture the feature interaction explicitly and then stacks MLP over the product layer to capture the high-order interaction implicitly.

- xDeepFM (Lian et al., 2018) also aims at capturing the high-order feature interaction explicitly layer by layer like DCN. But it uses a different feature interaction layer from DCN.

- GCN (Kipf & Welling, 2016) is a graph convolutional neural network whose goal is to capture the correlation among samples for prediction.

Throughout our experiments, we set the embedding size of each feature to 10 for all methods. As for the sample interaction component of our method, the dimension is set to $[16, 16]$. To make a fair comparison, GCN and the baseline methods with an MLP component are also set in the same way. As for the feature interaction component of our method, the dimension is set to $[10, 10, 10]$. Similarly, baseline methods with the high-order feature interaction component are also set to the same dimension. Moreover, the batch size is set to 1024 and the learning rate is set to 0.0001.

## 5.3 LINK PREDICTION

In Table 2, we show the results for link prediction across a number of graphs from different domains. In all cases, DRFM outperforms the baseline methods across all the different real-world data sets. Compared to the best baseline method, DRFM achieves a gain in AUC of 1.6% for Modcloth, 4.9% for Renttherunway, 0.4% for Book-crossing. From Table 2, we also observe that by combining FMs and GCNs, the standard deviation in AUC is much smaller than other FM-based methods and

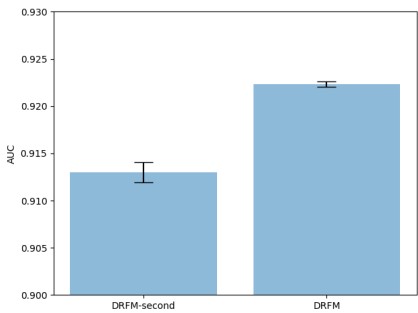

Figure 1: The link prediction result of DRFM and DRFM-second.

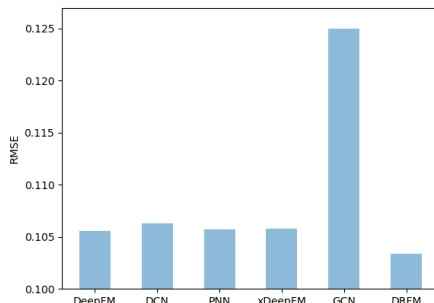

Figure 2: Regression results for Company-X-CTR.

typically less than GCN as well. These results indicate utility of combining FMs and GCNs for learning a more accurate and robust model for prediction.

To further verify the effectiveness of our proposed high-order feature interaction component, we compare DRFM with its variant which replaces the hihg-order feature interaction component with the regular FM component. Here, we call it DRFM-second since it can only capture the second-order feature interaction. Due to the space limitation, we only report the result of Modecloth in Figure 1. It can be seen that DRFM with our high-order feature interaction component has much better performance than that with only the regular FM component, which confirms the effectiveness of our high-order feature interaction component.

### 5.4 REGRESSION

To further verify the performance of our proposed method, we use DRFM to predict the winning rate of a campaign bidding for exposure. Here, each campaign is configured with different attributes, such as targeting countries, targeting device types, user segment rules, etc. Advertisers might change only one or two attributes and launch a another campaign. This new campaign and its original campaign share a lot of common information so that they are highly correlated. Thus, it will be beneficial to capture the relationship between different campaigns when making prediction. To this end, we construct the graph for campaigns. Specifically, in this dataset, this kind of new campaigns and their original campaigns share the same ID. Thus, we can construct the graph in terms of the shared ID. More specifically, two campaign are connected if they share a same ID.

The result is shown in Figure 2. Here, to measure the performance of different methods, we use Root Mean Squared Error (RMSE) as the metric. A smaller value indicates a better result. It can be seen that the regular GCN performs worse than all the other baseline methods. The possible reason is that GCN does not utilize the feature interaction for prediction, while feature interaction is confirmed to be a powerful technique in the high-dimensional CTR prediction. Moreover, our method DRFM outperforms all state-of-the-art FMs, which confirms the effectiveness of incorporating the relational information for prediction.

### 6 CONCLUSION

In this work, we described a new class of models that combine Factorization Machines (FMs) and Graph Neural Network (GNNs) into a unified learning approach. By seamlessly combining FMs and GNNs, we obtain the unique advantages offered by each while overcoming the issues that arise when either is used independently. Using real-world data from different domains, we demonstrated the effectiveness of combining GNNs and FMs for both link prediction and regression tasks. While this work demonstrated the utility of combining FMs and GNNs in a single unified learning framework, there remains many open research problems to investigate in future work. One important future direction is to explore other FM and GNN variants (besides the vanilla ones used in this work), and systematically investigate the utility and effectiveness of these different combinations.

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
