# OpenReview forum: "Deep Relational Factorization Machines"
_ICLR.cc/2020/Conference — Reject_

### Official Review · AnonReviewer2 · 2019-10-14
**Official Blind Review #2**

**Rating:** 1

**Review:**

In this paper, the authors propose generalize the FM to consider both interaction between features and interaction between samples. For the interaction between features, the authors propose to use graph convolution to capture high-order feature interactions. Moreover, the authors construct a graph on the instances based on similarity. Then a GCN is applied to the sample graph where the feature embedding is shared between the two components. Experiments are carried out on four datasets with tasks of link prediction and regression. Comparison to several baselines demonstrate the superior performance of the proposed method.

Strength:
1. The idea of utilizing GCN on the feature co-occurrence graph is interesting and innovative. The idea could possibly be combined with other variants of Deep FM models.
2. It is an interesting idea to combine sample similarity together with feature co-occurrence for better prediction accuracy.

Weakness:
1. Many descriptions in the paper are not very clear. First, the authors only mention how prediction is carried out with trained parameters. However, there is no description of the training process like what is the target used for the two components. What is the training procedure? Are the two components trained jointly? Second, the authors provide little description on how the sample similarity graph is constructed excepts for the Ad campaign dataset. Third, it is not clear how is the link prediction evaluation carried out. From the size of the graph, the authors seem to include both user and item in the graph. However, the user and item has disjointed feature set. It is not clear how the GCN is computed for the heterogenous nodes in the graph. Moreover, how is link prediction carried out, by taking inner product (cosine similarity) of the final representation.
2. For equation (8) in section 4.1, why we need to compute h_i^{RFI}. This should be the feature representation of sample i. However, the average is computed without include sample i itself. Also, are the neighbors defined in the sample similarity graph? Should we use the sample interaction in section 4.2 to capture that?
3. Though it is interesting idea to use graph convolution on the feature occurrence graph, it would be much better if the authors could provide more intuition on the output of the GCN. It would be helpful to study a few simple cases like without non-linearity. Is it a generalization to high-order FM without non-linearity? Also, it would be interesting to see experiments results using the graph convoluted feature representation directly for final representation. Also, some visualization of the learned feature embedding also helps.
4. The authors should carry out ablation study for different components of the model. Moreover, it would be much better if the authors can carry out experiments on some widely used recommendation datasets and use standard evaluation metrics for ranking.


**Experience Assessment:**

I have read many papers in this area.

**Review Assessment: Checking Correctness Of Derivations And Theory:**

I assessed the sensibility of the derivations and theory.

**Review Assessment: Checking Correctness Of Experiments:**

I assessed the sensibility of the experiments.

**Review Assessment: Thoroughness In Paper Reading:**

I read the paper at least twice and used my best judgement in assessing the paper.

---

### Official Review · AnonReviewer3 · 2019-10-23
**Official Blind Review #3**

**Rating:** 1

**Review:**

This paper proposes to combine the graph neural networks and factorization machines. First, the authors propose a relational feature interaction component (RFO) tp deal with the categorical features. This component first uses the factorization machine to project the features to h^FI(x), then it uses an aggregation operation to get the prediction y^RFI. To explore high-order correlations, the authors further propose to calculate a concurrence graph, on which RFI propagates the embedding vectors to get relational high-order correlations. To further model high-order sample interactions, this work then presents a special graph convolutional operation that considers the element-wise products of the encoded features.

My comments are as follows:

- The idea of integrating the GNN and FMs is interesting and intuitive. However, the proposed method is simple.

- As the work proposes a new model architecture, a graphical illustration and the pseudo-code is necessary for the audiences.

- Some parts are useless for the whole paper. For example, 'a straightforward method is to combine...' (Eq. 3). The discussion of the simple RFI-component is also needless since the paper mainly proposes the high-order version. These paragraphs should be simplified or removed.

- The graph convolution operation in Eq. (7) first considers all the element-wise products of the embedding vectors, which is the same as the original FM. Then the authors use G, the concurrence graph, to propagates the embedding vectors. One concern is that the original FMs also consider the graphical information in G, i.e. the concurrence relation. Will the GNN technique improve the usage of this topological information of the features?

- An ablation study is necessary to show the contribution of the proposals. For example, comparing the naive RFI and high-order RFI; the performance with and without RFI/SI components.

- I would be grateful if the authors provide the running time comparison of the proposed method.



**Experience Assessment:**

I have published one or two papers in this area.

**Review Assessment: Checking Correctness Of Derivations And Theory:**

I carefully checked the derivations and theory.

**Review Assessment: Checking Correctness Of Experiments:**

I carefully checked the experiments.

**Review Assessment: Thoroughness In Paper Reading:**

I read the paper thoroughly.

---

### Official Review · AnonReviewer1 · 2019-10-29
**Official Blind Review #1**

**Rating:** 3

**Review:**

This paper tries to combine FMs and GNNs to capture both sample and feature interactions. First, feed the feature from FMs to GNNs. Second, build high-order interactions.

Strength:
[1] This paper tries to solve an interesting question and the idea is simple and intuitive
[2] Experiments show that the proposed approach outperforms a number of baselines

My comments:
[1] main concern: lack of ablation study. It would be great to analyze the effects of different components and illustrate/visualize the learned features to see what's the difference and why/how such difference help
[2] another concern is complexity. It would be great to see learning curve and computational time analysis

**Experience Assessment:**

I do not know much about this area.

**Review Assessment: Checking Correctness Of Derivations And Theory:**

I assessed the sensibility of the derivations and theory.

**Review Assessment: Checking Correctness Of Experiments:**

I carefully checked the experiments.

**Review Assessment: Thoroughness In Paper Reading:**

I read the paper at least twice and used my best judgement in assessing the paper.

---

### Public Comment · ~Xiang_Wu1 · 2019-10-08
**Can you run your model on the Criteo dataset?**

It is a relatively old dataset, but it is much easier for us to understand your progress if you can provide numbers on this benchmark. The kaggle link is at:

https://www.kaggle.com/c/criteo-display-ad-challenge

---

> ### Author Response · Authors · 2019-10-23
> **The graph is not available for Criteo dataset.**
>
> Since we don't know the meaning of features of this dataset, we cannot construct the graph. If we know the meaning of features, we can construct the graph in terms of the practical property of features. For instance, in our Company-X-CTR data, each campaign is configured with different attributes, such as targeting countries, targeting device types, user segment rules, etc. Advertisers might change only one or two attributes and launch another campaign. This new campaign and its original campaign share a lot of common information so that they are highly correlated. Thus, it will be beneficial to capture the relationship between different campaigns when making prediction.  Specifically, this kind of new campaigns and their original campaigns share the same campaign_placement_id. Thus, we can construct the graph in terms of the shared campaign_placement_id. More specifically, two campaigns are connected if they share the same campaign_placement_id. After obtaining the graph, we can use our proposed method to make prediction.

---

### Decision · Program_Chairs · 2019-12-19

**Decision:**

Reject

**Comment:**

This paper proposes to combine FMs and GNNs. All reviewers voted reject, as the paper lacks experiments (eg ablation studies) and novelty. Writing can be significant improved - some information is missing. Authors did not respond to reviewers questions and concerns. For this reason, I recommend reject.